# The impact of national income and vaccine hesitancy on country-level COVID-19 vaccine uptake

Javad Moradpour[1], Ali Shajarizadeh[2], Jasmine Carter[3], Ayman Chit[1,4], Paul Grootendorst[1,5]*

1 Leslie Dan Faculty of Pharmacy, University of Toronto, Toronto, Ontario, Canada, 2 Independent Researcher, Vancouver, British Columbia, Canada, 3 Independent Researcher, Ottawa, Ontario, Canada, 4 Sanofi Ltd, Lyon, France, 5 Department of Economics, McMaster University, Hamilton, Ontario, Canada

* paul.grootendorst@utoronto.ca

## Abstract

### Background

The rapid development and rollout of COVID-19 vaccines helped reduce the pandemic's mortality burden. The vaccine rollout, however, has been uneven; it is well known that vaccination rates tend to be lower in lower income countries. Vaccine uptake, however, ultimately depends on the willingness of individuals to get vaccinated. If vaccine confidence is low, then uptake will be low, regardless of country income level. We investigated the impact on country-level COVID-19 vaccination rates of both national income and vaccine hesitancy.

### Methods

We estimated a linear regression model of COVID-19 vaccine uptake across 145 countries; this cross-sectional model was estimated at each of four time points: 6, 12, 18, and 24 months after the onset of global vaccine distribution. Vaccine uptake reflects the percentage of the population that had completed their primary vaccination series at the time point. Covariates include per capita GDP, an estimate of the percentage of country residents who strongly disagreed that vaccines are safe, and a variety of control variables. Next, we estimated these models of vaccine uptake by country income (countries below, and above the international median per capita GDP) to examine whether the impact of vaccine hesitancy varies by country income.

### Results

We find that GDP per capita has a pronounced impact on vaccine uptake at 6 months after global rollout. After controlling for other factors, there was a 22 percentage point difference in vaccination rates between the top 20% and the bottom 20% of countries ranked by per capita GDP; this difference grew to 38% by 12 months. The deleterious impact of distrust of vaccine safety on vaccine uptake became apparent by 12 months and then increased over time. At 24 months, there was a 17% difference in vaccination rates between the top 20% and the bottom 20% of countries ranked by distrust. The income stratified models reveal

**Data Availability Statement:** The data used to generate the results presented in the study are available in S2 Table.

**Funding:** PG acknowledges research seed funding support from the Leslie Dan Faculty of Pharmacy, University of Toronto. https://www.pharmacy.utoronto.ca. No grant number was assigned. JM acknowledges post-doctoral salary support from Mitacs Accelerate grant number IT23414. This award is funded by Sanofi Ltd. (50% of funding) and the Government of Canada and the Province of Ontario through the Ministry of Training, Colleges and Universities (50% of funding). JC acknowledges funding from the Leslie Dan Faculty of Pharmacy, University of Toronto. There was no additional external funding received for this study. The funders had no role in the design and conduct of the study; collection, management, analysis, and interpretation of the data; preparation, review, or approval of the manuscript; and decision to submit the manuscript for publication.

**Competing interests:** The authors have declared that no competing interests exist.

that the deleterious impact of vaccine distrust on vaccine uptake at 12 and 24 months is particularly evident in lower income countries.

## Conclusions

Our study highlights the important role of both national income and vaccine hesitancy in determining COVID-19 vaccine uptake globally. There is a need to increase the supply and distribution of pandemic vaccines to lower-income countries, and to take measures to improve vaccine confidence in these countries.

## 1. Introduction

Coronavirus disease-2019 (COVID-19) was declared a pandemic by the World Health Organization (WHO) in March 2020 [1]. Remarkably, the first COVID-19 vaccines were launched only nine months later, in December 2020 [2,3]. By July 2022, 12 billion COVID-19 vaccine doses had been administered globally and two thirds of the global population had received at least one dose [4]. The rapid roll-out of vaccines has thus markedly reduced the mortality and morbidity burden of the COVID-19 pandemic [5].

Vaccine uptake, however, has varied across countries. Auld and Toxvaerd [6] report that COVID-19 vaccine uptake at March 2021 was much higher in higher-income countries than in low income countries. Basak and colleagues report the same findings for vaccine uptake at December 2021 [7]. Ngo and colleagues find that higher income countries were able to vaccinate their residents faster than lower income countries were during 2021 [8].

Understanding how country income affects pandemic country vaccine coverage has implications for global vaccine policy. One reason is equity–ideally, those who bear the highest mortality or morbidity risk should be afforded the opportunity to get vaccinated regardless of their country of residence [9,10]. Another reason is efficiency—the duration and intensity of the global pandemic depends on the distribution of the vaccine across countries. One would want to vaccinate those who would otherwise pose the greatest risk of transmitting the virus, or its variants, regardless of where they reside. Global COVID-19 mortality, estimated at 6.8 million deaths as of February 2023, [4] could have been reduced had distribution been more efficient [11–14].

In this paper, we offer additional insights into the global roll-out of the COVID-19 pandemic vaccines. We do so by estimating linear regression models of country-level COVID-19 vaccine uptake at different points in time, focusing on the role of two factors: country income and vaccine hesitancy. Income is captured by per capita gross domestic product (GDP) and country population. GDP per capita is an important predictor of a country's ability to pay and acquire priority access to COVID vaccines, as is clear from the empirical evidence presented in the earlier studies [6,7]. It also likely reflects a country's ability to distribute vaccines to its residents. The total size of the market, measured using the country population, may also matter. It seems plausible that vaccine manufacturers may prioritize allocations to countries that offer larger total sales.

Country income alone, however, does not fully explain vaccine uptake. If there is widespread mistrust of vaccine safety and efficacy, or if individuals do not feel at risk of COVID-related morbidity or mortality [15], then vaccine uptake will be low, regardless of the ability of a country to procure and distribute vaccine doses. Thus, we also include in our vaccine uptake model covariates that reflect vaccine hesitancy.

The roles of country income and vaccine hesitancy likely vary over time. Six months after vaccines first became available, many countries, both low [16] and high income [17,18], struggled to procure enough vaccine for residents who were willing to get immunized. Thus, vaccine hesitancy likely played little, if any, role in determining COVID-19 vaccine uptake. Later, in high income countries at least, there were enough vaccines for everyone to get vaccinated. Vaccine hesitancy likely became the limiting factor.

We explored the relative impacts of country income and vaccine hesitancy over time by estimating four cross-sectional regression models of country level vaccine uptake, at 6, 12, 18 and 24 months after the start of the global roll-out of COVID-19 vaccines. We also estimate variants of the 12- and 24-month models in which the impact of vaccine hesitancy on vaccine uptake is permitted to vary by country income. This allows us to determine if hesitancy played a larger role in vaccine uptake in high income countries 12 and 24 months post global roll-out than in low-income countries.

## 2. Methods

Our outcome variable, $y_{it}$, is the percentage of residents of country $i$ who completed their primary COVID-19 vaccination series (2 doses for most vaccines, 1 or 3 for a few manufacturers) at the end of month $t$ ($t$ = June 2021, December 2021, June 2022, December 2022). Most of the vaccines given Emergency Use Listing status by the WHO at January 2022 require two doses. These include Pfizer/BioNTech (BNT162b2) [19], Oxford/AstraZeneca (ChAdOx1-S) [20], Moderna (mRNA-1273) [21], Sinopharm [22], Sinovac-CoronaVac [23], Bharat Biotech BBV152 COVAXIN [24], and Novavax [25]. The Janssen Ad26.COV2.S is the sole single-shot vaccine [26]. Completion of the primary vaccine series offers the strongest protection against serious illness, hospitalization and death [27]. Some other studies in this literature, by contrast, focus on the total number of doses (primary series and booster doses) administered per capita [6,8].

We used vaccine uptake and population data assembled by Our World in Data (OWD) to estimate $y_{it}$ [4]. OWD analysts use the vaccinations data periodically reported by governments, health ministries and other official public sources for 226 jurisdictions to generate daily cumulative estimates of $y_i$ [3]. OWD analysts validate the officially reported figures by checking for illogical values, implausible changes in vaccination rates and other irregularities. OWD analysts also receive feedback on data problems from their global user base [3].

We assumed that no vaccinations were provided in country $i$ prior to the date that $y_i$ was first reported by OWD. If data on $y_i$ was unavailable at the end of month $t$, but data were available for an earlier and a later date, we used an interpolated value. If there were no data reported after the end of month $t$, we used the most recent available estimate. In most cases, the estimate we relied on was reported earlier in the same month, typically within a week of the target date. One country in our analysis sample, the United Arab Emirates, had an apparent data error: the percentage of people who completed the initial vaccination protocol exceed 100% for the final two values of $t$. We replaced these values with 100%.

Country level covariates reflect year 2020 values, except as otherwise noted. Country level data on current dollar (i.e., not inflation adjusted) GDP per capita were obtained from the World Bank. National GDP was converted from the local currency into U.S. dollar (USD) equivalents using average annual market exchange rates. There were no 2020 World Bank GDP data for two jurisdictions in our analysis sample, Taiwan and Venezuela. We therefore imputed these missing values using the International Monetary Fund (IMF) estimates of 2020 current USD per capita GDP. In particular, we estimated a linear regression model of the World Bank GDP series with the IMF GDP series as the sole covariate. Using this model, we

predicted the World Bank values using the IMF 2020 estimates for Taiwan and Venezuela. Data on country population, both totals and by age group, were obtained from the United Nations 2022 Revision of its World Population Prospects report [28].

Country level vaccine hesitancy was measured using several covariates. The primary covariate is the estimated percentage of residents who, as of December 2019, strongly disagreed with the statement that vaccines are safe. Data come from de Figueiredo and colleagues [29], who summarized country-wide confidence about vaccine safety and efficacy using data from 290 surveys administered in 149 countries between September 2015 and December 2019. We focused on confidence in vaccine safety. Although people's opinion about the importance or effectiveness of vaccines may play a role in the uptake of other vaccines, concern about side effects of COVID-19 vaccines was the most common reason for hesitancy [29,30].

de Figueiredo and colleagues estimate country-specific vaccine hesitancy three months prior to the official start of the COVID-19 pandemic. Ideally, one would instead use COVID vaccine hesitancy estimated around each of the four time points (June 2021, December 2021, June 2022, December 2022) that we analyse. However, these data are unavailable. Lazarus and colleagues [31] produced estimates of COVID vaccine confidence for 23 jurisdictions, but not for all the countries in our sample.

We controlled for vaccine hesitancy using two additional covariates, country level educational attainment and the elderly (age 65+) share of the population. There is evidence that more highly educated individuals tend to be more forward looking, and thus more willing to invest for their future [32]. Thus, countries with higher levels of educational attainment may have higher levels of vaccine uptake. There is also strong evidence that COVID-related mortality increases sharply with age after age 65 [33]. Thus the elderly may be less concerned about any adverse effects of the COVID-19 vaccine than COVID-19 disease itself. High death rates among the elderly may also prompt others to become vaccinated [15]. Thus one would expect higher vaccination rates in countries with a larger elderly share of the population; the elderly population share was estimated using the United Nations population data.

The United Nations Development Program (UNDP) produces two widely-used measures of country level education. One is expected years of schooling, measured as the number of years a child of school entrance age is expected to spend in school; this is estimated using current elementary, secondary and tertiary school enrollment rates. Ngo and colleagues [8] found this variable to be an important predictor of the number of COVID-19 vaccine doses per capita in a country. The UNDP also produces estimates of the mean years of schooling of the population aged 25 years and older. We preferred the latter measure; the actual educational attainment of adults presumably better reflects levels of vaccine acceptance in the population. The measure used by Ngo, which reflects school enrollment rates for those under age 25, appears to capture public and private investment in a country's education system.

The UNDP does not produce schooling attainment estimates for Taiwan. We thus imputed the value for Taiwan using a linear regression model of 2020 UNDP mean years of schooling for those aged 25+. The sole covariate in this model was a forecast of country level mean years of schooling among individuals aged 25–64 years for 2020, produced by Barro and Lee [34]. The model predicted an educational attainment for Taiwan, given the Barro and Lee 2020 forecast for Taiwan, of 12.9 years; this predicted value is the 16th highest in our country dataset.

In addition to these market size and vaccine hesitancy covariates, we controlled for the geographic region in which the country is located: the United States, Canada and Western Europe; Eastern Europe and Central Asia; South Asia; East Asia and the Pacific; Middle East and North Africa; Sub-Saharan Africa; and Latin America and the Caribbean. East Asia and the Pacific formed the reference category. These locational indicators are intended to capture additional factors that affect the capacity of the country to procure and distribute vaccines, the risk of

COVID-related mortality perceived by inhabitants and other factors that affect the willingness of residents to become vaccinated.

In an earlier version of this paper [35] we also controlled for population weighted density estimated by the University of Southampton [36]. While this variable has been shown to be an important predictor of vaccine uptake in other studies [37,38], it had no material effects on vaccine coverage in our models after we controlled for GDP, beliefs about vaccine safety, educational attainment, and the other covariates. Ngo and colleagues similarly found that a standard population density measure (residents per squared kilometre) had no discernable impact in their models. This variable was therefore removed from the model.

Our analysis sample consists of 145 countries; the population of these countries constituted over 97% of the world population in 2020. These 145 countries include all but four of the jurisdictions for which de Figueiredo and colleagues' [29] vaccine confidence estimates were available. (The OWD group reported no data on the percentage of the population of North Cyprus, Kosovo or Switzerland that had completed the primary vaccination series; these data were missing for Luxembourg for 26 consecutive months of our analysis period.) The data sources are provided in S1 Table; the data itself are provided in S2 Table.

When positing the structure of a linear regression model of a fractional outcome, such as the vaccinated share of the population, some analysts log-transform continuous covariates [6]. We used a different functional form. For each continuous covariate, we grouped countries in the analysis sample into 5 quintiles and represented group membership with 4 indicator variables. The bottom 20% of countries for each covariate formed the reference group in each case. For instance, the indicator for the top GDP quintile equals one for the 20% of countries with the highest per capita GDP. The regression model parameter for this indicator represents the difference in vaccine uptake relative to the 20% of countries with the lowest per capita GDP. The use of quintile indicators allows the vaccine coverage model more flexibility in fitting the data than does the log transform. Moreover, the use of quintile indicators mitigates the impact of covariate measurement error when the mismeasured and actual covariate values are in the same quintile.

We re-estimated the regression model for vaccine uptake at 12 and at 24 months post global roll-out, allowing the model parameters to vary by GDP per capita. To do so, we estimate the models separately for the countries in the bottom and top 50% of countries by GDP per capita. To conserve degrees of freedom, we use the log of the per capita GDP, vaccine confidence, and expected years of schooling covariates, instead of their quantiles. The parameters for these covariates reflect the impact of a one percent (proportional) increase in the covariate on the vaccinated share of the population.

We estimated the parameters of the regression models using ordinary least squares (OLS) as implemented in Stata version 17 [39]. Parameter estimates were graphed using the coefplot Stata program [40]. Because the error term variance may vary across observations (i.e., the errors might be heteroskedastic), we estimated the OLS standard errors using the heteroskedasticity-robust covariance matrix estimator. We also estimated the estimator precision of the income-stratified models using the wild bootstrap estimator [41] as this estimator has been shown to be more reliable in smaller samples. Ethics approval was not needed given that this study relies on aggregated, national-level data.

## 3. Results

Table 1 presents summary statistics for the outcome and explanatory variables across the 145 countries in the analysis sample. The mean percentage of the country population that completed their primary COVID-19 vaccinations increased from 14% at June 2021, to 43% in

**Table 1. Summary statistics for outcome and explanatory variables.**

| Variable | Mean | SD | Min | Max |
|---|---|---|---|---|
| % Fully vaccinated (June 30, 2021) | 13.72 | 16.48 | 0.00 | 65.31 |
| % Fully vaccinated (Dec. 31, 2021) | 42.80 | 27.14 | 0.00 | 96.40 |
| % Fully vaccinated (June 30, 2022) | 50.60 | 26.79 | 0.11 | 100.00 |
| % Fully vaccinated (Dec. 31, 2022) | 53.63 | 25.51 | 0.23 | 100.00 |
| GDP per capita | 13,384 | 17,809 | 217 | 85,420 |
| Population (thousands) | 52,697 | 169,540 | 367 | 1,424,930 |
| % Strongly disagree vaccine is safe | 3.26 | 2.5 | 0.5 | 17.5 |
| Mean years of schooling among 25+ | 13.83 | 3.0 | 7.0 | 21.1 |
| % Age 65 or older | 9.77 | 6.9 | 1.7 | 29.6 |
| Geographic region: | | | | |
| East Asia and the Pacific | 13% | 0.34 | 0 | 1 |
| Eastern Europe and Central Asia | 20% | 0.40 | 0 | 1 |
| Latin America and the Caribbean | 13% | 0.34 | 0 | 1 |
| Middle East and North Africa | 12% | 0.33 | 0 | 1 |
| South Asia | 4% | 0.20 | 0 | 1 |
| Sub-Saharan Africa | 25% | 0.43 | 0 | 1 |
| USA, Canada and Western Europe | 12% | 0.33 | 0 | 1 |

*n = 145; SD = standard deviation; GDP = gross domestic product; USA = United States of America.*

December 2021, and to 54% in December 2022. The estimated standard deviation for the four vaccine uptake variables was highest for the December 2021 measure. At that time, vaccine uptake varied from 0% in Ethiopia to 96% in the United Arab Emirates.

Per capita GDP varies markedly across countries, from the equivalent of $217 USD in Burundi to $85,420 USD in Ireland. Population size also varies widely, from 366,669 (Iceland) to 1.4 billion (China). The percentage of the population that strongly disagrees that vaccines are safe varies from 0.46% in Thailand to 17.49% in Azerbaijan; the mean is 3.3%. The average years of schooling among those 25+ years varies from 2.1 years in Burkina Faso to 14.1 years in Germany. The elderly share of the population varies from 1.7% in the United Arab Emirates to 29.6% in Japan.

Fig 1 displays country level vaccination rates at December 31, 2022, by the percentage of the population that strongly disagrees that vaccines are safe. Each panel highlights in blue observations on countries in a different income quintile and fits a spline curve to these observations. The data indicate that vaccination rates tend to be lower, the lower is per capita GDP. Also, within an income quintile, vaccination rates tend to be lower in countries with a greater degree of vaccine hesitancy.

Fig 2 graphs the estimated parameters and 95% confidence intervals of regression models of the percentage of population that completed their primary COVID-19 vaccinations as of June 30, 2021, Dec. 31, 2021, June 30, 2022, and Dec. 31, 2022. S3 Table reports, for each regression model estimated in this paper, i) the parameter estimates, ii) estimated standard errors, iii) t-statistics and p-values associated with the hypothesis test that each parameter equals zero, iv) 95% confidence intervals and v) adjusted R squared values.

We first examine the role of GDP per capita in explaining vaccine uptake at successive dates after the initial global vaccine roll-out. At 6 months post roll-out (June 30, 2021), we find a gradient in vaccination rates by GDP quintile. Countries in the top 20% of the distribution of per capita GDP have vaccinated an additional 22% of their residents compared to countries in

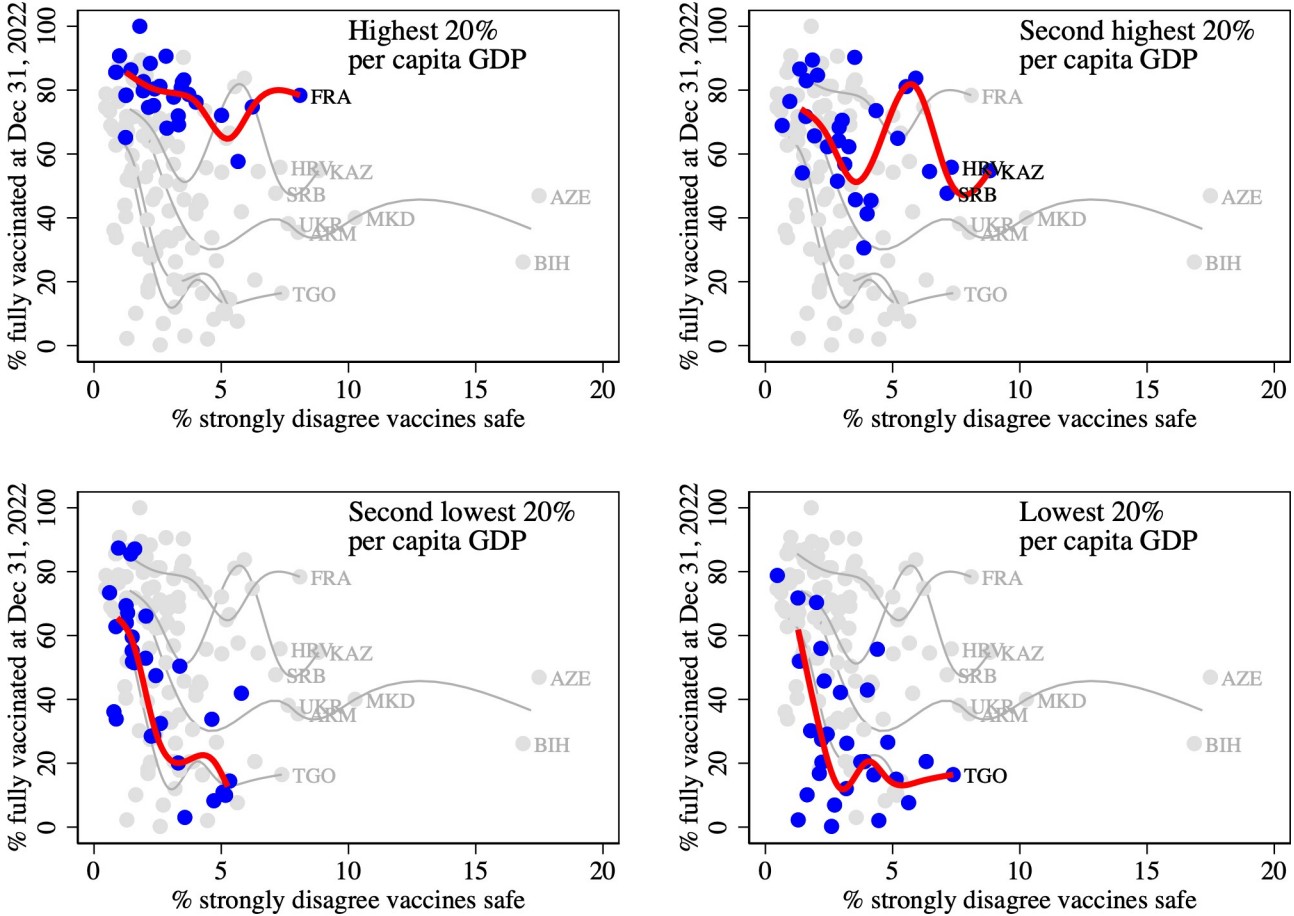

**Fig 1. Vaccine uptake at December 31, 2022, by % of population that strongly disagrees that vaccines are safe.** Note: Each panel highlights observations on countries in a different income quintile and fits a spline curve to these observations. Countries in which more than 7% of the population strongly disagrees that vaccines are safe are labelled. FRA = France, HRV = Croatia, SRB = Serbia, UKR = Ukraine, ARM = Armenia, AZE = Azerbaijan, BIH = Bosnia and Herzegovina, KAZ = Kazakhstan, MKD = North Macedonia, TGO = Togo.

the bottom quintile. The vaccination rate difference for the countries in the second highest quintile and those at the bottom are 17%. Countries in the second and third quintile have only slightly higher (about 3%) vaccine uptake than those in the lowest quintile group; these differences are not statistically significant at conventional levels.

The country income-vaccination gradient increases in the 12-month model (Dec. 31, 2021). At 12 months into the rollout of the COVID vaccines, the countries in the top 20% of the distribution of per capita GDP had vaccinated an additional 38% of their residents compared to countries in the bottom quintile. Countries in the second quintile have vaccinated an additional 31% of their residents compared to countries in the bottom quintile.

The difference in vaccine uptake between the countries in the top two income quintiles and those in the bottom quintile decrease slightly during 2022. This may reflect growth in the vaccine uptake of the bottom quintile; as evidence, the estimated value of the constant term increases from 3% at 6 months post roll-out to 49% at 24 months post roll-out.

The impact of population size on vaccine uptake also varies over time. At 6 months post roll-out, the most populous countries have vaccination rates that are about 7% lower than those in the smallest countries. The population-size-related differences in vaccine uptake, however, largely disappear over time. The magnitude of the estimated differences between

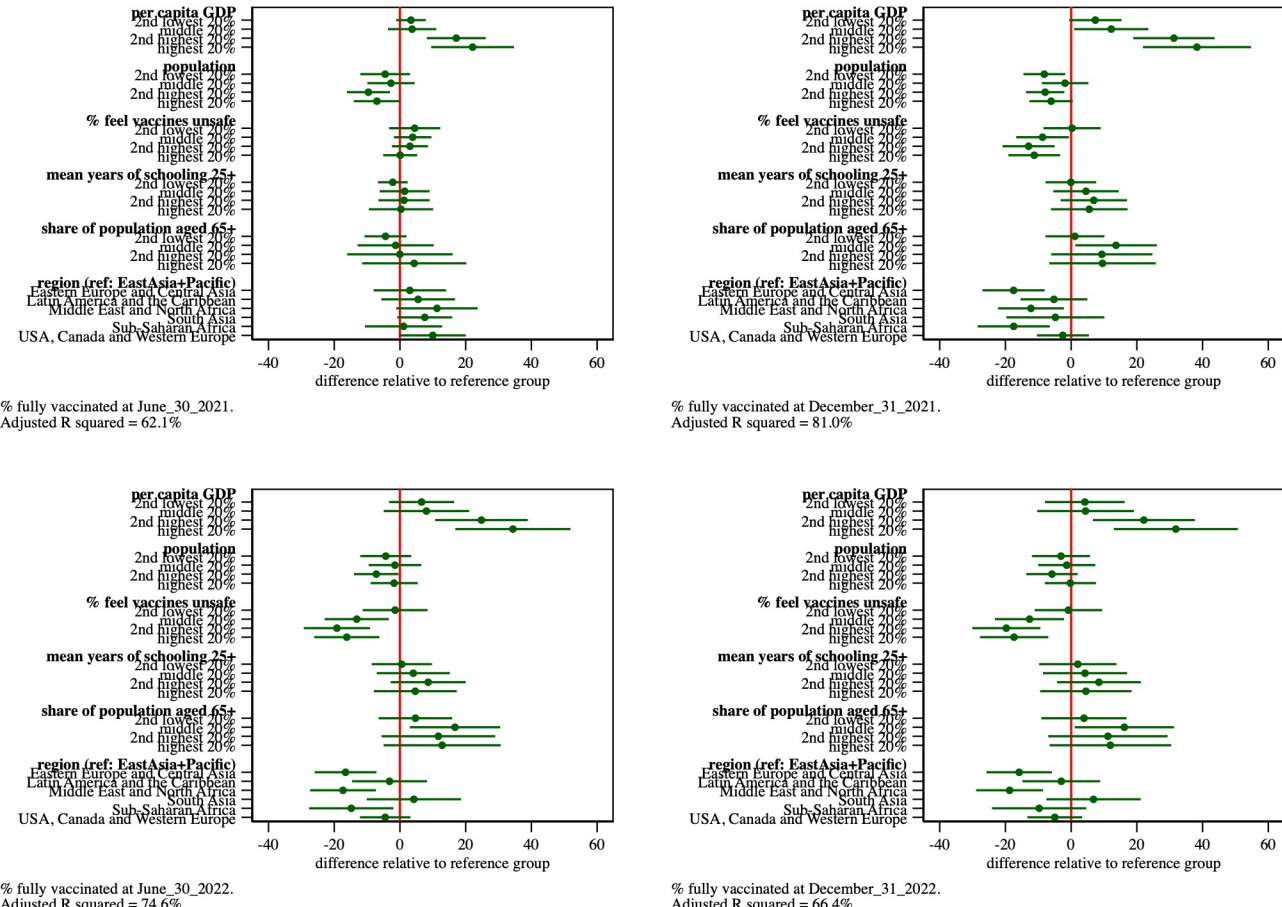

**Fig 2. Estimated parameters and 95% confidence intervals from regression model of percentage of population that completed primary COVID-19 vaccinations as of June 30, 2021, Dec. 31, 2021, June 30, 2022, Dec. 31, 2022, respectively.**

population quintiles are much smaller and not significant at conventional levels in the 18- and 24-month models. The 6-month results may thus simply reflect the fact that it takes longer to vaccinate a large country than a smaller country.

We turn next to the impact on vaccine uptake of vaccine hesitancy, as measured by the percentage of the population that, just prior to the COVID-19 pandemic, strongly disagreed with the statement that vaccines are safe. Recall the covariate is measured using indicators of its 5 quintiles, with the indicator of the fifth quintile capturing the 20% of countries with the highest share of the population who mistrust vaccine safety. These indicators have little impact on vaccination rates 6 months post roll-out. By 12 months post roll-out, however, the impact of vaccine hesitancy is evident in the estimates. Countries in the fourth and fifth quintiles of vaccine hesitancy have vaccination rates that are 13 and 11 percentage points, respectively, lower than countries in the first quintile. Countries in middle quintile have vaccination rates about 9% lower than countries in the first quintile. The vaccination rates difference between the top and bottom quintiles grows to about 17% by 24 months.

The mean years of schooling among those 25 and older, a measure of vaccine acceptance, had negligible effects on vaccine uptake at 6 months. By 12 months, an educational gradient in vaccine uptake emerges; the magnitudes of the estimates, however, are small and not statistically significant at conventional levels.

The estimated impact of the elderly share of the population on vaccination rates had mixed effects on vaccine uptake. Compared to the 20% of countries with the smallest elderly shares, countries in the second to fifth quintiles had higher rates of vaccine uptake. There was no apparent gradient in estimates, however; estimates were also largely insignificant at conventional levels.

Countries located in Eastern Europe and Central Asia, and countries in the Middle East and North Africa had persistently lower rates of vaccine uptake, after controlling for GDP, vaccine hesitancy and other covariates. At 24 months post global roll-out, the Eastern Europe and Central Asia countries had about an 16% lower vaccination rate, and the Middle East and North Africa countries had a 19% lower vaccination rate compared to countries in the reference group (East Asia and the Pacific).

The model fit, measured by the adjusted R squared, increases from 62% in the 6-month model, to 81% in the 12-month model. The adjusted R squared then declines to 75% in the 18-month model and to 66% in the 24-month model. This indicates that the national income, vaccine hesitancy, and the remaining covariates explain most of the international variation in vaccine uptake around 12 months after first global vaccine roll-out.

Taken together, our models suggest that country per capita GDP–which reflects the ability of a country to purchase and distribute vaccines–accounts for a 38 percentage point difference in country vaccine uptake about one year into the global vaccine rollout. GDP declines in declines in importance thereafter. At the same time, the share of the population that harbours concerns over vaccine safety becomes a binding constraint on vaccine uptake. Having controlled for GDP and confidence in vaccine safety, the mean years of schooling and the elderly share of the population have relatively small effects on vaccine uptake.

## 3.1 Sensitivity analyses

We estimated variants of the model to assess the sensitivity of the results to different model specifications. We first re-estimated the model after removing the sets of covariates that had little apparent impact on vaccine uptake: population size, mean years of schooling, and the elderly share of the population. Results appear in Table 3 of S3 Table and are graphed in Fig 3. The removal of these covariates has little impact on the estimated impact of vaccine confidence on vaccine uptake. The country GDP-vaccine uptake gradient, however, increases. Per capita GDP now accounts for a 48 percentage point difference in country vaccine uptake about one year into the global vaccine rollout. Adjusted R-squares decrease by 1–2 percentage points.

Next, we added to this model expected years of schooling, a covariate used in Ngo's vaccine uptake models. Results appear in Table 4 of S3 Table and are graphed in Fig 4. The inclusion of this covariate has no material impact on the estimated effects of vaccine confidence. However, the covariate reduces the magnitude of the impact of per capita GDP on vaccine uptake. Adjusted R-square values increase by 4–5 percentage points in the 12, 18 and 24 month models.

## 3.2 Subgroup analyses

We re-estimated the regression model for vaccine uptake at 12 and at 24 months post global roll-out, allowing the model parameters to vary by GDP per capita. To do so, we estimate the models separately for the countries in the bottom and top 50% of countries by per capita GDP.

The parameter estimates, graphed in Fig 5 and displayed in Table 5 of S3 Table, suggest that the impact of vaccine hesitancy (measured as the share of the population that strongly mistrusts vaccine safety) varies markedly between subgroups. In the higher-income subgroup vaccine hesitancy explains little of the country difference in vaccine uptake at 12 and 24 months. In the lower-income subgroup, however, each one percent (proportional) increase in vaccine

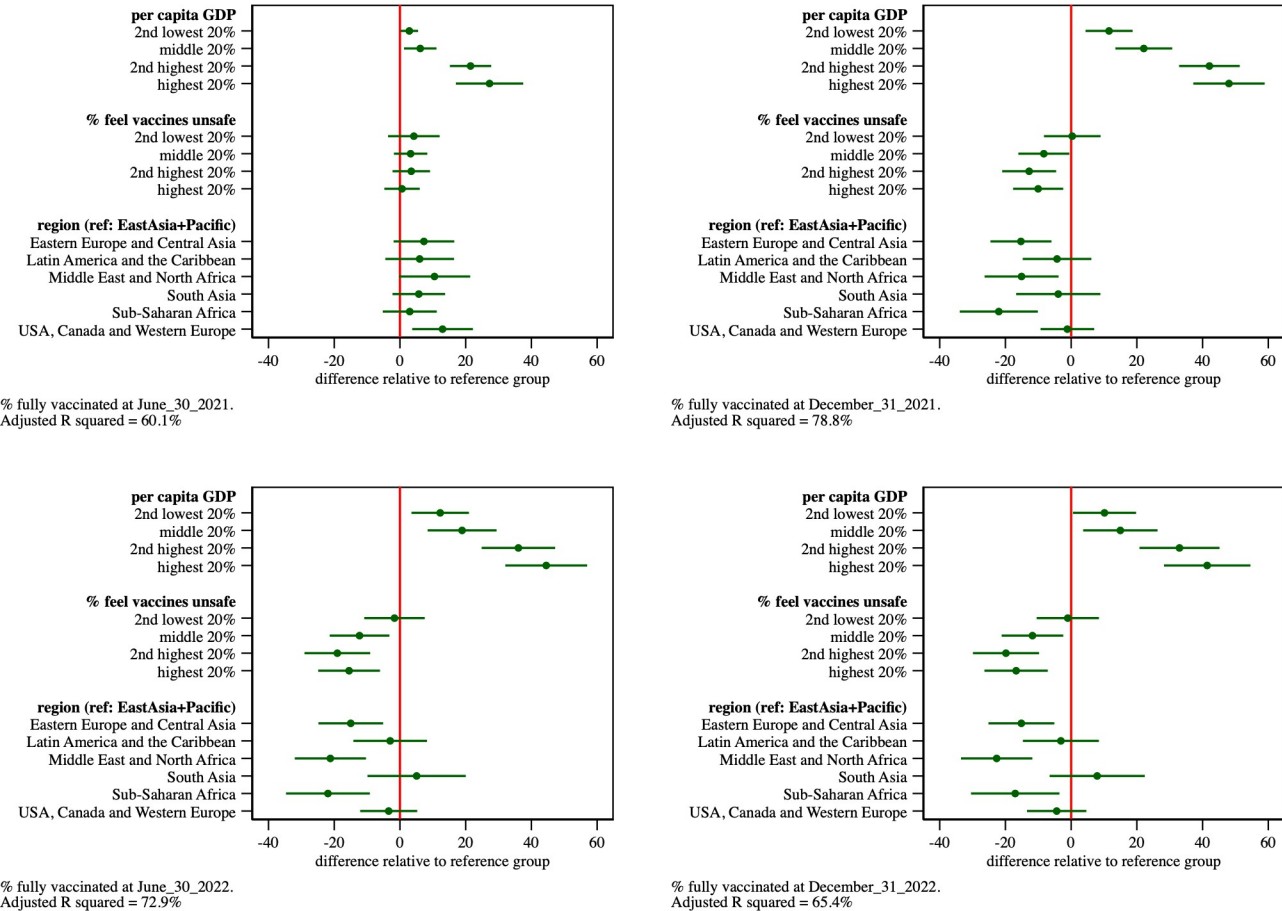

**Fig 3. Estimated parameters and 95% confidence intervals from regression model of percentage of population that completed primary COVID-19 vaccinations as of June 30, 2021, Dec. 31, 2021, June 30, 2022, Dec. 31, 2022, respectively.** Population size, mean years of schooling, and the elderly share of the population covariates removed from model.

distrust decreases vaccine uptake at December 31, 2021 by about 9 percentage points. The magnitude of this effect grows to about 18 percentage points at December 31, 2022.

The estimates also reveal that the geographic differences in vaccine uptake vary by country income. In particular, the relatively low vaccine uptake in the 18 Middle East and North African countries observed earlier appears to be limited to the subset of these countries that are in the lower income subgroup.

We re-estimated the p-values for these income-specific regression models using the wild bootstrap estimator as implemented for Stata by Roodman and colleagues [41]. (This procedure generates the bootstrapped distribution of robust t-test statistics using 999 replications, with Rademacher weights and with the restriction that the parameter is equal to zero imposed.) The bootstrapped p-values, which appear in Table 6 of S3 Table, are consistent with the p-values reported in Table 5.

## 4. Discussion

This paper explores how national income and vaccine hesitancy affected COVID-19 vaccine uptake across 145 countries after vaccines became available globally in December 2020. We estimate regression models of the share of the population that completed their primary

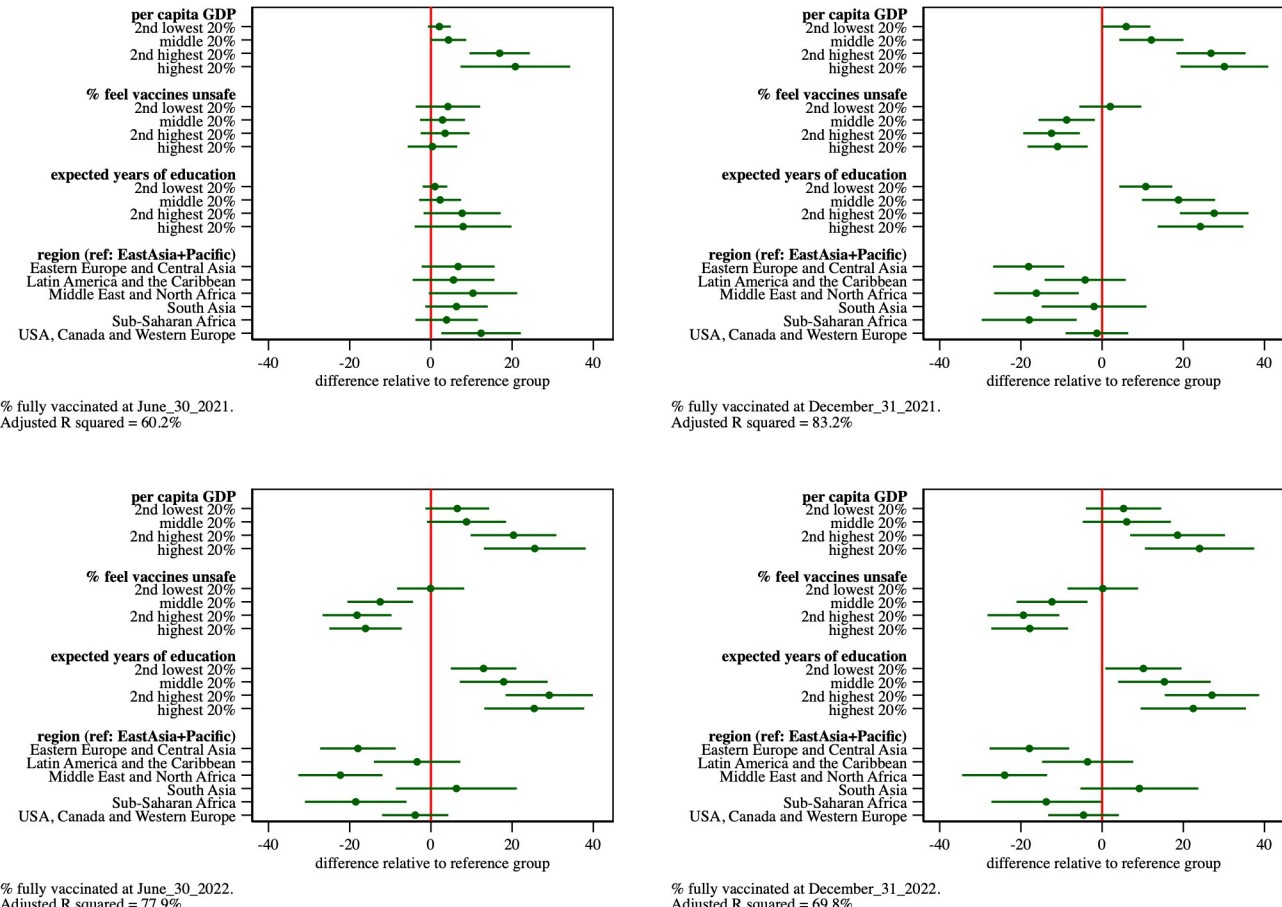

**Fig 4. Estimated parameters and 95% confidence intervals from regression model of percentage of population that completed primary COVID-19 vaccinations as of June 30, 2021, Dec. 31, 2021, June 30, 2022, Dec. 31, 2022, respectively.** Population size, mean years of schooling, and the elderly share of the population covariates removed from model. Expected years of schooling included in model.

vaccination series at 6-, 12-, 18- and 24-months post vaccine rollout. The primary covariates are per capita GDP and the fraction of the population that mistrusts vaccine safety; control variables include population size, the age 65+ share of the population, levels of educational attainment and indicators for different geographic regions.

We find marked income-related differences in vaccine take-up at 6 months after vaccines became available. The model R squared was relatively low, however. This suggests that factors outside the model played a particularly important role in determining vaccine uptake in the short term. Rosen, Waitzberg and Israeli [42] examine some of these factors in the context of Israel's vaccine uptake. They attribute Israel's rapid vaccine uptake to, *inter alia*, "well-developed infrastructure for implementing prompt responses to large-scale national emergencies", and the "organizational, IT and logistical capacities of Israel's community-based health care providers." There are undoubtedly other unique factors that explain the short run vaccine uptake in other countries.

The greatest income-related differences in vaccine uptake are observed at 12 months into the global rollout of the COVID-19 vaccines; the GDP-vaccine uptake gradient declines thereafter. This decline may reflect an expansion of supply, which would tend to lower vaccine prices; it may also reflect the donation of vaccines to lower income countries. Cross country income differences, however, still account for a 32 percentage point difference in our model of vaccine uptake across countries at 24 months.

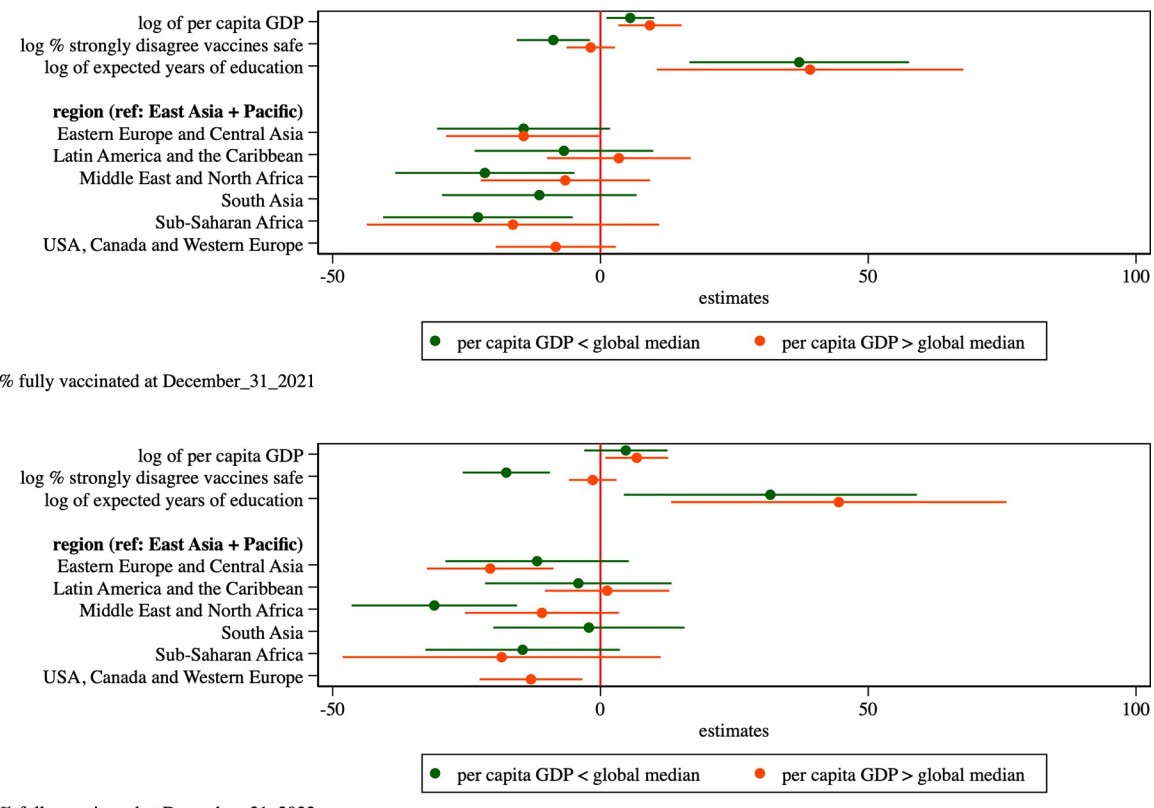

% fully vaccinated at December_31_2021

% fully vaccinated at December_31_2022

**Fig 5. Estimated parameters and 95% confidence intervals from regression model of percentage of population that completed primary COVID-19 vaccinations as of Dec. 31, 2021, and Dec. 31, 2022, respectively.** Estimates provided separately for countries in the bottom and top 50% of countries by GDP per capita. The parameters for the log of per capita GDP, vaccine confidence, and expected years of schooling covariates reflect the impact of a one percent (proportional) increase in the covariate on the vaccinated share of the population.

The estimated impact of GDP on vaccine uptake depends on the choice of control variables. For instance, when using just geographic region indicator controls, we observe a 48 percentage point difference in 12 month uptake between the top and bottom 20% of countries ranked by per capita GDP. When a measure of educational attainment, expected years of schooling, is then added to the model, this uptake difference declines to 30 percentage points and model fit improves considerably. The expected years of schooling, used by another study in the literature, reflects enrollment rates in primary, secondary and tertiary education; it does not directly measure the average level of education of the adult population. Thus, it does not seem to directly reflect vaccine acceptance. Instead, the predictive power of this variable appears to be due to its ability to capture the capacity of a country to vaccinate its population.

Regardless of the choice of control variables, the models find that country-level vaccine uptake is lower, the greater the fraction of the population that mistrusts vaccine safety. This effect is first observed in the 12-month models and grows in magnitude thereafter. This makes sense. At 6 months, countries struggled to procure sufficient vaccine doses for all those willing to get vaccinated. Later, in high income countries at least, there were enough vaccines for everyone to get vaccinated. Vaccine hesitancy likely became the limiting factor.

To further investigate, we estimated the 12- and 24-month regression models separately by country income levels–the bottom and the top 50% of countries ranked by per capita GDP–to allow for income-related differences in the impact of vaccine hesitancy on the percentage of

the population fully vaccinated. We hypothesized that hesitancy would play a larger role in high income countries given that at 24 months post global roll-out, all residents of high-income countries who wanted to be fully vaccinated would have done so, leaving just the vaccine hesitant unvaccinated. Conversely, in low-income countries, even at 24 months not all individuals who were willing to get vaccinated would be, so that vaccine procurement and distribution capacity, not hesitancy, would be the constraining factor.

We found that the opposite was true. Hesitancy played a larger role in determining vaccine uptake in low-income countries than in high-income countries. This result is consistent with findings by Dayton and colleagues [43], who conducted phone surveys with more than 120,000 respondents in 53 low- and middle-income countries; these surveys were conducted between October 2020 and August 2021. The authors find that about 20% of adults in these countries were hesitant about the COVID-19 vaccine, with the most cited reason for hesitancy being concerns about vaccine safety. Hesitancy may be one of the reasons that age-specific COVID-19 related mortality rates were roughly 2 times higher in lower income countries than in higher income countries [44].

Our results are tempered by a limitation common to regression models estimated with observational data, namely, the possibility that the mean of the error term distribution varies with covariate values. This can occur in several ways. First, it is possible that there are omitted covariates, captured in the error term, that are correlated with either per capita GDP or confidence in vaccine safety. The fact that our primary results did not depend on the choice of control variables provides at least some assurance that omitted variables bias is not a problem.

Second, there may be measurement error in key covariates. This is a possibility given that the measure of public confidence in vaccine safety was estimated from survey data. There is also evidence that the GDP figures by authoritarian governments are upwards biased [45]. Moreover, the population and GDP data for 2020 could have been subject to some error given that the collection of these data was disrupted in many countries due to the COVID pandemic. Our choice to model the quantiles of the covariate, instead of the covariate itself, can mitigate the consequences of this measurement error.

Third, there could be simultaneity bias: the covariate values could be determined by the outcome variable values. This does not seem to be a problem since all covariates in our models were measured in 2020 or earlier, before the outcome (vaccine uptake) was determined. There might also be some measurement error in the outcome variable, vaccine uptake. The OWD group performs validity checks on the official vaccination figures produced by country governments, but OWD does not audit these data.

With these caveats in mind, we conclude that both per capita GDP and mistrust of vaccine safety have important effects on COVID-19 vaccine uptake beginning at one year after the global vaccine roll-out. The deleterious impact of mistrust of vaccine safety on vaccine uptake, however, appears to be most pronounced in lower-income countries. Initiatives to increase the supply and distribution of pandemic vaccines to lower-income countries, and to take measures to bolster vaccine confidence in these countries is thus required to improve vaccine coverage in these countries. Doing so will pay dividends in improving the effectiveness of vaccines in ameliorating future pandemics.

## Supporting information

**S1 Table. Sources of the variables that appear in the regression models.**
(XLSX)

**S2 Table. Data used to estimate the regression models.**
(XLSX)

**S3 Table. Estimated regression models of percentage of population fully vaccinated at 6, 12, 18 and 24 months post global roll-out.**
(XLSX)

## Author Contributions

**Conceptualization:** Javad Moradpour, Ayman Chit, Paul Grootendorst.

**Data curation:** Javad Moradpour, Ali Shajarizadeh, Jasmine Carter, Paul Grootendorst.

**Formal analysis:** Javad Moradpour, Ali Shajarizadeh, Paul Grootendorst.

**Funding acquisition:** Ayman Chit.

**Investigation:** Jasmine Carter.

**Methodology:** Javad Moradpour, Paul Grootendorst.

**Supervision:** Ali Shajarizadeh, Ayman Chit, Paul Grootendorst.

**Visualization:** Paul Grootendorst.

**Writing – original draft:** Javad Moradpour, Ali Shajarizadeh, Paul Grootendorst.

**Writing – review & editing:** Javad Moradpour, Jasmine Carter, Ayman Chit, Paul Grootendorst.

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
