## [Decision Letter · Decision Letter 0]

10 Jul 2023

PONE-D-23-10138The impact of national income and vaccine hesitancy on country-level COVID-19 vaccine uptakePLOS ONE

Dear Dr. Grootendorst,

Thank you for submitting your manuscript to PLOS ONE. After careful consideration, we feel that it has merit but does not fully meet PLOS ONE’s publication criteria as it currently stands. Therefore, we invite you to submit a revised version of the manuscript that addresses the points raised during the review process.

We look forward to receiving your revised manuscript.

Kind regards,

Engidaw Fentahun Enyew, MSc

Academic Editor

PLOS ONE

Journal Requirements:

"PG acknowledges research support from the Leslie Dan Faculty of Pharmacy, University of Toronto. https://www.pharmacy.utoronto.ca

No grant number was assigned."

Reviewers' comments:

Reviewer's Responses to Questions

**Comments to the Author**

1. Is the manuscript technically sound, and do the data support the conclusions?

Reviewer #1: Yes

Reviewer #2: No

Reviewer #3: Yes

2. Has the statistical analysis been performed appropriately and rigorously? 

Reviewer #1: Yes

Reviewer #2: No

Reviewer #3: Yes

3. Have the authors made all data underlying the findings in their manuscript fully available?

Reviewer #1: Yes

Reviewer #2: No

Reviewer #3: Yes

4. Is the manuscript presented in an intelligible fashion and written in standard English?

Reviewer #1: No

Reviewer #2: No

Reviewer #3: Yes

5. Review Comments to the Author

Reviewer #1: Thank you for the opportunity to review the manuscript titled “The impact of national income and vaccine hesitancy on country level COVID-19 vaccine up”

You did a good job. It will have a great contribution to the community. Saying this, I do have some concerns that could have been addressed.

Comments

Introduction

1. The mortality rate of the pandemic in developing countries (especially, in the Sub-Saharan countries) was not as high as those in developed countries. I would expect a narration under the introduction regarding how much the difference was in terms of cases, mortality, and morbidity between the two extreme regions (developed and developing ones). That is because every individual may not be needed to be vaccinated, certain groups of the population could be naturally protected and we all are not the same.

2. In your introduction section, “Most vaccines have been found to reduce infection and mortality rates”. Does that mean some vaccines do not affect the virus? If so, how could an individual be confident enough to take the vaccines?

Method

1. “Strongly disagreed with the statement that vaccines are safe” was the question you used to measure vaccine hesitancy. Do you think vaccine hesitancy could have an equivalent measurement before and after COVID-19? And how?

2. It would be better if you put an operational for hesitancy for COVID-19 vaccine uptake including the measurement you used.

Reviewer #2: Obviously, the research topic is dealing with a very important issue, that is challenged by every country in the globe, however I am strongly believe that the approach that the researchers were handled this topic is insufficient in term of originally or significance, that prevent / reject it from being published for the following reasons:

• Failing to include key variables that are affecting in vaccine uptake that were included in previous.

• Insufficient, inadequate and confusion in describing the dependent and independent variables.

• Insufficient and inadequate literature review.

• I believe that studying the impact of vaccine hesitancy on COVID-19 vaccine uptake can be studied more accurately and precisely in each single country rather than 147 countries.

• Statements of hypothesis without any evidence as in the Method section line #38 “We hypothesized that individuals with more years of education and older adults are more accepting of vaccines”.

• The researchers rely on the data from other studies e.g. [Our World in Data (OWD), International Monetary Fund (IMF), World Bank estimate for 2020, United Nations 2022 Revision of its World Population Prospects report, de Figueiredo and colleagues, Wittgenstein Centre for Demography and Global Human Capital, population weighted density estimated by the University of Southampton], these instruments were not enough described, in term of strengths and weakness, as well as psychometric properties.

• The results are inadequate and disorganized, for example the first table should provide a description for each country by population, GDP, density, educational level….etc. furthermore the tables are confusing, not clear, disorganized and concise, also they are hard to understand and interpret.

• Although the title of the figure1 is present, but the figure itself is absent.

• One of the factors that the researchers focusing on is the role of vaccine hesitancy, this factor was not discussed and explained sufficiently.

• The Statistical analysis are not fully described.

Thank you,

Reviewer #3: 1. This is a very well written Manuscript.

2. There are additional recommendations for possible corrections.

a. Authors should number their lines.

b. At methods, change ‘THe’ to "The" on page 3

d. The authors statement “These include the percentage of the population over 25 years who have post-secondary education ….” This statement excludes the age 25. However, the statement "The percentage of the population 25+

with post-secondary education...." on page 7 line 7, includes age 25. Also check table 1. Kindly correct.

e. Correct “25+”. Should it have been “ 25 years +”? Check page 7, line 7. Correct same in Table 1.

f. Did authors indicate the software used for the statistical analysis?

Thank you.

6. PLOS authors have the option to publish the peer review history of their article (what does this mean?). If published, this will include your full peer review and any attached files.

Reviewer #1: No

Reviewer #2: **Yes: **Dr. Mohammad Othman Abudari

Reviewer #3: **Yes: **Addae Hammond

---

## [Author Response · Author response to Decision Letter 0]

23 Aug 2023

Reviewer #1: Thank you for the opportunity to review the manuscript titled “The impact of national income and vaccine hesitancy on country level COVID-19 vaccine up”

You did a good job. It will have a great contribution to the community. Saying this, I do have some concerns that could have been addressed.

Response:

We thank the referee for the conscientious review of the paper and the kind words. 

Comments Introduction

1. The mortality rate of the pandemic in developing countries (especially, in the Sub-Saharan countries) was not as high as those in developed countries. I would expect a narration under the introduction regarding how much the difference was in terms of cases, mortality, and morbidity between the two extreme regions (developed and developing ones). That is because every individual may not be needed to be vaccinated, certain groups of the population could be naturally protected and we all are not the same.

Response:

Our reading of the literature is that rates of age-specific COVID-19 related mortality were markedly higher in lower income countries than in higher income countries. For a meta analysis of the evidence, see Levin et al BMJ Global Health 2022

https://gh.bmj.com/content/7/5/e008477.long

Nevertheless, we agree with the referee’s point that individuals’ perceptions of their COVID mortality risk will affect their demand for vaccinations. In particular, those who feel that they have natural immunity against COVID might decline the vaccination.

We make this point clear in the methods section where we note that the geographic region indicators will capture variation in vaccine uptake due to variation in perceived COVID mortality risk:

In addition to these market size and vaccine hesitancy covariates, we controlled for the geographic region in which the country is located: the United States, Canada and Western Europe; Eastern Europe and Central Asia; South Asia; East Asia and the Pacific; Middle East and North Africa; Sub-Saharan Africa; and Latin America and the Caribbean. East Asia and the Pacific formed the reference category. These locational indicators are intended to capture additional factors that affect the capacity of the country to procure and distribute vaccines, the risk of COVID related mortality perceived by inhabitants and other factors that affect the willingness of residents to become vaccinated. 

2. In your introduction section, “Most vaccines have been found to reduce infection and mortality rates”. Does that mean some vaccines do not affect the virus? If so, how could an individual be confident enough to take the vaccines?

The referee makes a very good point. It is possible that uptake in some countries was low because residents understood that the vaccines approved for use in the country were not particularly effective or safe. We return to this point in a moment. 

Re the statement “most vaccines have been found to reduce infection and mortality rates”: it was not our intention here to emphasize the differences in effectiveness of the different covid vaccines. What we meant to say is that within months of the start of the pandemic, there were effective vaccines introduced that reduced infection and mortality risk. Thus, the rapid uptake of these vaccines reduced mortality rates globally. To remove this distraction, we removed the statement (and associated references) from the introduction. 

Method

1. “Strongly disagreed with the statement that vaccines are safe” was the question you used to measure vaccine hesitancy. Do you think vaccine hesitancy could have an equivalent measurement before and after COVID-19? And how?

The referee asks whether confidence in COVID vaccines, in particular, could explain some of the variation in vaccine uptake. We agree that it would be preferable to use data on country level COVID-vaccine-specific confidence and we now make this point clear in the paper at line 153. Ideally the confidence measures would be measured at multiple points in time given that vaccine hesitancy likely varied over time within a country. We suspect that when people saw that their acquaintances received COVID vaccinations without significant adverse events, then they felt more comfortable to get vaccinated themselves. So, one would need multiple snapshots on hesitancy for the same country. Unfortunately, these data are simply not available. 

There is some limited information, however. Lazarus and colleagues (Nature Medicine, 2023) produced estimates of COVID vaccine confidence for 23 jurisdictions, but not for all the countries in our sample.

In this Nature study, vaccine hesitancy was defined as having reported ‘no’ to the question of whether they have received at least one dose of a COVID-19 vaccine and also ‘unsure/no opinion’, ‘somewhat disagree’ or ‘strongly disagree’ to the question of whether they would take a COVID-19 vaccine when available to them. 

https://www.nature.com/articles/s41591-022-02185-4

Thus the Nature measure is a less stringent definition of hesitancy than the ones we use. Nevertheless, it is noteworthy that in the set of countries for which both Lancet and Nature estimates were available, our Lancet measure predicts the Nature measure with an R squared of about 18%.

. regress hesitantpercent safe_strongly_disagree_2019

 Source | SS df MS Number of obs = 23

-------------+---------------------------------- F(1, 21) = 4.47

 Model | 641.760187 1 641.760187 Prob > F = 0.0467

 Residual | 3017.8746 21 143.708314 R-squared = 0.1754

-------------+---------------------------------- Adj R-squared = 0.1361

 Total | 3659.63478 22 166.347036 Root MSE = 11.988

 hesitantpercent | Coefficient Std. err. t P>|t| [95% conf. interval]

+----------------------------------------------------------------

safe_strongly_disagree_2019 | 3.160854 1.49575 2.11 0.047 .0502717 6.271435

 _cons | 12.04621 4.86282 2.48 0.022 1.933423 22.159

2. It would be better if you put an operational for hesitancy for COVID-19 vaccine uptake including the measurement you used.

As we noted above, we agree that a post COVID measure would be preferred. We attempted to create such a measure; we measured the number of WHO Emergency Use Listing (EUL)-designated COVID vaccines approved in each country and at each of the four time points, June 30, 2021, Dec 31, 2021, June 30, 2022, and Dec 31, 2022. Presumably, COVID vaccines that have the WHO EUL-designation would be more trusted than vaccines that did not get the WHO seal of approval. We included this covariate alongside the Lancet measure of vaccine hesitancy and found that it had no material effects on vaccine uptake. 

Reviewer #2: Obviously, the research topic is dealing with a very important issue, that is challenged by every country in the globe, however I am strongly believe that the approach that the researchers were handled this topic is insufficient in term of originally or significance, that prevent / reject it from being published for the following reasons:

We thank the referee for the feedback; we are sorry to hear that the referee found our study to be both unoriginal and insignificant. 

• Failing to include key variables that are affecting in vaccine uptake that were included in previous.

Among other criticisms, the referee suggests that our model fails to capture certain unnamed key variables that were included in previous studies.

We acknowledge that there are factors that affect country level COVID vaccine uptake that were not captured using our model covariates. This is clear since the model R squared values are less than 100%. We note, however, that our goal is to estimate the effect of per capita GDP and vaccine hesitancy on vaccine uptake. The inclusion of additional covariates in our model is only necessary to the extent that the covariate affects vaccine uptake and is correlated with the covariates of interest. Including such covariates reduces the deleterious effect of confounding (i.e. omitted variables bias). We are not aware of any covariates that have been used in the extant literature on country level COVID vaccine uptake that are potential confounders. 

• Insufficient, inadequate and confusion in describing the dependent and independent variables.

It is difficult to respond to the referee’s concerns absent any specific examples. We reviewed the text and simply could not see how the description of the variables in our model is insufficient, inadequate or confusing.

• Insufficient and inadequate literature review.

In light of the referee’s criticism, we performed our literature search again; this included a search of all papers that have cited the studies we cited in our original manuscript. This search yielded a relevant new study that modeled COVID vaccination rates as a function of GDP and other covariates. 

Ngo, V.M., Zimmermann, K.F., Nguyen, P.V. et al. How education and GDP drive the COVID-19 vaccination campaign. Arch Public Health 80, 171 (2022). 

The study, conducted by Ngo and colleagues, examined country level vaccine uptake during 2021, the first year in which vaccines were available. We now include it in our literature review. We thank the referee for the helpful suggestion to re-do the literature review.

• I believe that studying the impact of vaccine hesitancy on COVID-19 vaccine uptake can be studied more accurately and precisely in each single country rather than 147 countries.

It is unclear to us how one can estimate the impact of hesitancy on vaccine uptake separately for each single country, let alone get a precise estimate. 

To begin, one cannot estimate our linear regression model using a single observation so presumably the referee proposes that we analyse different data.

The referee did not describe the proposed method or data to improve estimator precision, so all we can do is speculate. One possibility is that the referee is requesting that we estimate our linear regression model separately for each country, using the daily or perhaps weekly time series observations on vaccine uptake per country. But this would not work since the vaccine hesitancy and other country level covariates in our model do not vary over time and would thus be collinear.

Another possibility is that the referee proposes that we estimate a different model using country specific individual level cross sectional (or longitudinal) data. But we are not aware of such data. To our knowledge there is no database that measures, across all our study countries, individual-level uptake of COVID vaccines and individual characteristics using a consistently defined set of variables. Even if we could find survey data for specific region, the results would be limited to that region and not necessarily generalizable to the rest of the world. 

• Statements of hypothesis without any evidence as in the Method section line #38 “We hypothesized that individuals with more years of education and older adults are more accepting of vaccines”.

This is a good point. We now motivate the inclusion of educational attainment and the elderly share of the population as covariates in our model by citing the relevant literature.

• The researchers rely on the data from other studies e.g. [Our World in Data (OWD), International Monetary Fund (IMF), World Bank estimate for 2020, United Nations 2022 Revision of its World Population Prospects report, de Figueiredo and colleagues, Wittgenstein Centre for Demography and Global Human Capital, population weighted density estimated by the University of Southampton], these instruments were not enough described, in term of strengths and weakness, as well as psychometric properties.

There is a lot to unpack here. 

First, our model uses country level demographic and socio-economic aggregates, like total population, educational attainment and gross domestic product. These variables are commonly used in this literature and in other literature that relies on cross country comparisons. 

When the referee requests that we assess the strengths and weakness and psychometric properties of country population counts, GDP and the other variables, we interpret this to mean that the referee is asking that we check whether the variables get at the underlying construct of interest (have construct validity) and are measured with minimal measurement error (are reliable).

To address the reviewer’s request, below we assess the validity and reliability of each of the variables in the model. Note that we have revised our model in light of the evidence presented by Ngo and colleagues. We also note that we have dropped the population density variable given that it was a weak predictor of vaccine uptake in both our original model and the models presented in Ngo et al. Our original model also used the fraction of the population with a tertiary educational degree; these data were estimated by the Wittgenstein Centre for Demography and Global Human Capital. Ngo et al use the UNDP education measures and now we do as well to enhance comparability. With this said, we now turn to the validity and reliability of each of the variables in the model. 

Vaccination coverage: % of the population which has obtained their primary vaccination series. This measure has good validity – it measures what we want and we cite a review study that finds that the completion of the primary vaccination series offers the greatest protection against severe illness. We note here that the extant literature – the papers by Auld, Basak, and Ngo – use the same vaccine uptake data source, i.e., OWD. We also now include in the manuscript a description of the validation checks that the OWD analysts perform on the vaccination data provided by governments. We acknowledge in the limitations section of the paper that the government data OWD relies on are not audited and thus subject to some measurement error.

Per capita GDP measured in US dollars is the standard measure of a country’s ability to purchase vaccines on international markets. The per capita GDP series in the original manuscript was measured by the IMF. Most studies, however, rely on the World Bank series. We thus use the World Bank series in the revised manuscript to enhance comparability of our results to those of other studies. There will no doubt be some measurement error in these GDP series and indeed in the other covariates in our model. We acknowledge this in the limitations section of the paper. We minimize the effect of measurement error by grouping countries into quintiles of per capita GDP and using indicators of these quintiles as covariates. The other continuous covariates are handled in the same manner. As we note in the paper, the use of quintile indicators mitigates the impact of mismeasurement in the covariate when the mismeasured and actual covariate values are in the same quintile.

Population: these are produced by the United Nations Population division. This group uses data from the civil registration and vital statistics systems, population censuses, population registers and household surveys of the UN member countries. The UN population data are the reference standard source of population for many countries. That being said, we acknowledge in the limitation section that the population figures for 2020 could have been subject to some error given that the collection of population data was disrupted in many countries due to the COVID pandemic.

Vaccine confidence: as we noted earlier, the de Figueiredo and colleagues Lancet data on confidence in vaccine safety is the best data available. It was measured a few months before the official start of the COVID-19 pandemic and it appears to have good content validity. But these data are estimates which will be measured with error. We acknowledge this in the limitations section of the paper. 

Education: as noted above we now use the UNDP educational estimates. We note here that we use the UNDP estimates of the mean years of schooling for adults aged 25 years and older and in some model specifications, the expected years of schooling for a person of school entrance age. The UNDP data came recommended by Ngo and colleagues. The education measures are included given that more highly educated are more forward looking and those who are more forward looking presumably are more willing to get vaccinated. Unfortunately, we could find no evidence on the reliability of the UNDP education estimates. That being said, as we demonstrate in the new sensitivity analysis section of the revised manuscript, (section 3.1 starting at line 355) the estimates of the impact of vaccine hesitancy on vaccine uptake are largely invariant to the inclusion of the educational attainment variables. 

• The results are inadequate and disorganized, for example the first table should provide a description for each country by population, GDP, density, educational level….etc. furthermore the tables are confusing, not clear, disorganized and concise, also they are hard to understand and interpret.

The referee recommends that we include as the first table in the results section data on the model outcome and covariate values for each of the 145 countries separately. This would be unconventional. Normally one presents a table of summary statistics in health services research and economics papers. However, in light of the referee’s request we now provide the entire analysis data set as a supplementary file (S2_Table). We also provide a table that describes our data sources and their URLs (S1_Table).

The referee next indicates that the tables of regression parameter estimates are hard to understand and interpret. We agree. Thus, in the revised manuscript we display the estimates and confidence intervals graphically. These are much easier to interpret. The regression estimates, standard errors, t-stats and p values are contained in S3_Table.

• Although the title of the figure1 is present, but the figure itself is absent.

The figures are provided in auxiliary files, as required by PLOS One.

• One of the factors that the researchers focusing on is the role of vaccine hesitancy, this factor was not discussed and explained sufficiently.

The referee has raised a valid point. We have expanded our discussion of vaccine hesitancy in the paper. Please see our responses to Referee #1

• The Statistical analysis are not fully described.

It is unclear to us exactly what aspect of the statistical analysis the referee found to be not fully described. We described our analysis dataset; the outcome and covariates of our regression model, the method to estimate the model parameters (OLS) and the methods to estimate estimator precision.

Reviewer #3: 1. This is a very well written Manuscript.

We thank the referee for the conscientious review of the paper and the kind words. We are happy to hear that the referee found the manuscript to be very well written.

2. There are additional recommendations for possible corrections.

a. Authors should number their lines.

We have made this change.

b. At methods, change ‘THe’ to "The" on page 3

We have corrected this typo; thanks for catching this.

d. The authors statement “These include the percentage of the population over 25 years who have post-secondary education ….” This statement excludes the age 25. However, the statement "The percentage of the population 25+

with post-secondary education...." on page 7 line 7, includes age 25. Also check table 1. Kindly correct.

e. Correct “25+”. Should it have been “ 25 years +”? Check page 7, line 7. Correct same in Table 1.

Thanks for pointing these typos out. We have fixed them. 

f. Did authors indicate the software used for the statistical analysis?

We now indicate that we used Stata v 17 for the statistical analysis.

Thank you.

---

## [Decision Letter · Decision Letter 1]

8 Oct 2023

The impact of national income and vaccine hesitancy on country-level COVID-19 vaccine uptake

PONE-D-23-10138R1

Dear Dr. Grootendorst,

We’re pleased to inform you that your manuscript has been judged scientifically suitable for publication and will be formally accepted for publication once it meets all outstanding technical requirements.

Kind regards,

Clifford Afoakwah, Ph.D.

Academic Editor

PLOS ONE

Additional Editor Comments (optional):

Reviewers' comments:

Reviewer's Responses to Questions

**Comments to the Author**

1. If the authors have adequately addressed your comments raised in a previous round of review and you feel that this manuscript is now acceptable for publication, you may indicate that here to bypass the “Comments to the Author” section, enter your conflict of interest statement in the “Confidential to Editor” section, and submit your "Accept" recommendation.

Reviewer #1: All comments have been addressed

Reviewer #3: All comments have been addressed

2. Is the manuscript technically sound, and do the data support the conclusions?

Reviewer #1: Yes

Reviewer #3: Yes

3. Has the statistical analysis been performed appropriately and rigorously? 

Reviewer #1: Yes

Reviewer #3: Yes

4. Have the authors made all data underlying the findings in their manuscript fully available?

Reviewer #1: Yes

Reviewer #3: Yes

5. Is the manuscript presented in an intelligible fashion and written in standard English?

Reviewer #1: Yes

Reviewer #3: Yes

6. Review Comments to the Author

Reviewer #1: Dear Author, I would like to appreciate your effort to address the issues that have been raised in the previous review

Reviewer #3: (No Response)

7. PLOS authors have the option to publish the peer review history of their article (what does this mean?). If published, this will include your full peer review and any attached files.

Reviewer #1: No

Reviewer #3: **Yes: **Addae Yaw Hammond

---

## [Editor Report · Acceptance letter]

23 Oct 2023

PONE-D-23-10138R1 

The impact of national income and vaccine hesitancy on country-level COVID-19 vaccine uptake 

Dear Dr. Grootendorst:

I'm pleased to inform you that your manuscript has been deemed suitable for publication in PLOS ONE. Congratulations! Your manuscript is now with our production department. 

Kind regards, 

on behalf of

Dr. Clifford Afoakwah 

Academic Editor

PLOS ONE